# Decision-Focused Learning without Differentiable Optimization: Learning Locally Optimized Decision Losses

**Sanket Shah**
Harvard University
sanketshah@g.harvard.edu

**Kai Wang**
Harvard University
kaiwang@g.harvard.edu

**Bryan Wilder**
Carnegie Mellon University
bwilder@andrew.cmu.edu

**Andrew Perrault**
The Ohio State University
perrault.17@osu.edu

**Milind Tambe**
Harvard University
milind_tambe@harvard.edu

## Abstract

Decision-Focused Learning (DFL) is a paradigm for tailoring a predictive model to a downstream optimization task that uses its predictions in order to perform better *on that specific task*. The main technical challenge associated with DFL is that it requires being able to differentiate through the optimization problem, which is difficult due to discontinuous solutions and other challenges. Past work has largely gotten around this this issue by *handcrafting* task-specific surrogates to the original optimization problem that provide informative gradients when differentiated through. However, the need to handcraft surrogates for each new task limits the usability of DFL. In addition, there are often no guarantees about the convexity of the resulting surrogates and, as a result, training a predictive model using them can lead to inferior local optima. In this paper, we do away with surrogates altogether and instead *learn* loss functions that capture task-specific information. To the best of our knowledge, ours is the first approach that entirely replaces the optimization component of decision-focused learning with a loss that is automatically learned. Our approach (a) only requires access to a black-box oracle that can solve the optimization problem and is thus *generalizable*, and (b) can be *convex by construction* and so can be easily optimized over. We evaluate our approach on three resource allocation problems from the literature and find that our approach outperforms learning without taking into account task-structure in all three domains, and even hand-crafted surrogates from the literature.

## 1   Introduction

Predict-then-optimize [7, 8] is a framework for using machine learning to perform decision-making. As the name suggests, it proceeds in two stages—first, a predictive model takes as input *features* and makes some *predictions* using them, then second, these predictions are used to parameterize an optimization problem that outputs a *decision*. A large number of real-world applications involve both prediction and optimization components and can be framed as predict-then-optimize problems—for e.g., recommender systems in which missing user-item ratings need to be predicted [13], portfolio optimization in which future performance needs to be predicted [17], or strategic decision-making in which the adversary behavior needs to be predicted [14].

In addition to wide applicability, this framework also formalizes the relationship between prediction and decision-making. This is important because such predictive models are typically learned inde-

36th Conference on Neural Information Processing Systems (NeurIPS 2022).

pendently of the downstream optimization task in Machine Learning-based decision-making systems, and recent work in the predict-then-optimize setting [8, 16, 25, 9, 3, 25, 27, 26, 6] has shown that it is possible to achieve *better task-specific performance* by tailoring the predictive model to the downstream task. This is often done by differentiating through the entire prediction and optimization pipeline end-to-end, leading to a family of approaches that we will refer to as *decision-focused learning (DFL)* [25]. Optimizing directly for the quality of decisions induced by the predictive model in this end-to-end manner yields a loss function we call the *decision loss*.

However, training with the decision loss can be challenging because the solutions to optimization problems are often discontinuous in the predictions (see Section 2.1). This results in an uninformative loss function with zero or undefined gradients, neither of which are useful for learning a predictive model. To address this, DFL approaches often leverage handcrafted surrogate optimization tasks that provides more useful gradients. These surrogate problems may be constructed by relaxing the original problem [8, 16, 25, 9], adding regularization to the objective [3, 25, 27], or even using entirely an different optimization problem that shares the same decision space [26].

Designing good surrogates is an art, requiring manual effort, insight into the optimization problem of interest. In addition, there are no guarantees that the surrogates induce convex decision losses, leading to local optima that further complicate training. Instead, we propose a fundamentally different approach: to *learn* a decision loss directly for a given task, circumventing surrogate problem design entirely. Our framework represents the loss as a function in a particular parametric family and selects parameters which provide an informative loss for the optimization task. We call the resulting loss a *locally optimized decision loss (LODL)*.

Our starting point is the observation that a good decision loss should satisfy 3 properties: it should (i) be *faithful* to the original task, i.e., the decision quality is consistent with the original problem; (ii) provide informative gradients everywhere (i.e., defined and non-zero); and (iii) be convex in prediction space to avoid local minima. These demands are in tension—the first requirement prevents the loss function from being modified too much to achieve the other two. It is not obvious apriori that any tractable parametric family should be able to simultaneously satisfy all three properties for the complex structure induced by many optimization tasks. We resolve this tension by separately modeling the loss function *locally* for the neighborhood around each individual training example. Faithful representation of the decision loss is easier to accomplish locally in each individual neighborhood than globally across instances, allowing us to introduce convex parametric families of loss functions which capture structural intuitions about properties important for optimization. To fit the parameters, we sample points in the neighborhood of the true labels, evaluate the decision loss associated with these sampled points, and then train a loss function to mimic the decision loss.

We evaluate LODLs on three resource allocation domains from the literature [12, 25, 24]. Perhaps surprisingly, we find that LODLs outperform handcrafted surrogates in two out of the three. In our analysis, we discover a linear correlation between the agreement of the learned LODL with the decision loss and the decision loss of a predictive model learned using said LODL. Our approach motivates a new line of research on decision-focused learning.

## 2 Background

In predict-then-optimize, a predictive model $M_{\boldsymbol{\theta}}$ first takes as input features $\boldsymbol{x}$ and produces predictions $\hat{\boldsymbol{y}} = M_{\boldsymbol{\theta}}(\boldsymbol{x})$. These predictions $\hat{\boldsymbol{y}}$ are then used to parameterize an optimization problem that is solved to yield decisions $\boldsymbol{z}^*(\hat{\boldsymbol{y}})$:

$$\boldsymbol{z}^*(\hat{\boldsymbol{y}}) = \arg\min_{\boldsymbol{z}} \ f(\boldsymbol{z}; \hat{\boldsymbol{y}})$$
$$s.t. \ \ g_i(\boldsymbol{z}) \leq 0, \text{ for } i \in \{1, \ldots, m\} \tag{1}$$

Note that, unlike typical machine learning problems, the dimensionality of $\dim(\hat{\boldsymbol{y}}) = D$ is likely to be *large* as it consists of all predictions needed to parameterize the optimization problem. Given the large dimensionality, and the similarity in the role of all the predictions, the predictive model typically predicts individual components of $\hat{\boldsymbol{y}}$, i.e. $\hat{\boldsymbol{y}} = [\hat{y}_1, \ldots, \hat{y}_D] = [M_{\boldsymbol{\theta}}(x_1), \ldots, M_{\boldsymbol{\theta}}(x_D)]$.

Predictions are evaluated with respect to the *decision loss (DL)* of the decision that they induce, i.e., the value under the objective function of the optimization under the ground truth parameters $\boldsymbol{y}$:

$$DL(\hat{\boldsymbol{y}}, \boldsymbol{y}) = f(\boldsymbol{z}^*(\hat{\boldsymbol{y}}), \boldsymbol{y})$$

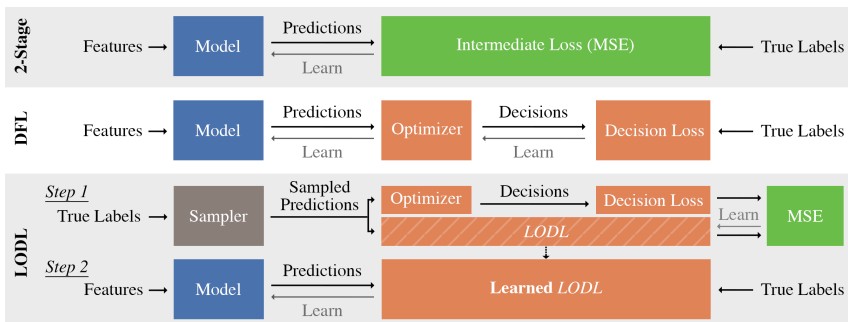

Figure 1: Schematic highlighting how the predictive model $M_\theta$ is learned in different approaches. LODL does not require backpropagating through the optimization problem.

Thus, for a dataset $[(x_1, y_1), \ldots, (x_N, y_N)]$, we aim to learn a model $M_\theta$ that generates predictions $[\hat{y}_1, \ldots, \hat{y}_N]$ that minimize the decision loss $DL$:

$$\theta^* = \arg\min_{\theta} \frac{1}{N} \sum_{n=1}^{N} DL(M_\theta(x_n), y_n)$$

This is in contrast with standard supervised learning approaches in which the quality of a prediction is measured by a somewhat arbitrary *intermediate loss* (e.g., mean squared error) that does not contain information about the downstream decision-making task. In this paper, we refer to models that use an intermediate loss as *2-stage* and those that directly optimize for $DL$ as *decision-focused learning (DFL)*. Figure 1 outlines how a predictive model is learned using these different approaches.

## 2.1 Motivating Example

Consider an $\arg\min$ optimization where the goal is to predict the *dis*utility $y$ of 2 agents (A, B), e.g., $y = (0, 1)$. Now, if these parameters are predicted perfectly, the decision is $z =$ "Pick A", and the decision loss $DL$ is the true disutility of agent A, i.e., $DL = 0$.

On the other hand, consider the set of predictions $\hat{y}_{bad} = (1 \pm \epsilon_A, 0 \pm \epsilon_B)$ for $0 < \epsilon_A, \epsilon_B < 0.5$. Any prediction in this set will yield the decision 'Pick B" and a decision loss $DL$ of 1. Given that the decision loss is constant in this region, the gradients are all zero, i.e., $\nabla_{\hat{y}} DL(\hat{y}, y)\big|_{\hat{y} \in \hat{y}_{good}} = 0$. As a result, if a predictive model makes such a prediction, it cannot improve its predictions by gradient descent. Therefore, although $DL$ is what we want to minimize, we *don't want to fit it perfectly*.

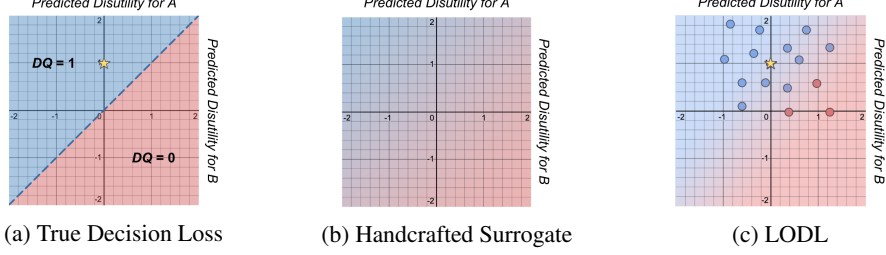

(a) True Decision Loss      (b) Handcrafted Surrogate      (c) LODL

Figure 2: Graphs plotting the values of different loss functions (Blue = 1, Red = 0) as a function of the predictions of different disutilities for A and B.

If instead of minimizing $DL$ directly, we minimized a surrogate loss $SL = \hat{y}_A - \hat{y}_B$, gradient descent would lead to the model to predict $\hat{y} = (-\infty, \infty)$ regardless of the initialization (see Figure 2). While $\hat{y}$ isn't the *true* set of disutilities, it does lead to the optimal decision, i.e., "Pick A" and is thus effective from a predict-then-optimize perspective.

The challenge is then coming up with such surrogates. Although it is easy for the problem above, it becomes more complicated as the number and types of variables and constraints grows. Below, we propose an approach to *automatically learn* good surrogates. Figure 2c shows what our method looks like for the example above.

# 3   Related Work

A great deal of recent work on decision-focused learning and related topics aims to incorporate information about a downstream optimization problem into the training of a machine learning models. Some optimization problems (especially strongly convex ones) can be directly differentiated through [7, 2, 1]. For others, particularly discrete optimization problems with ill-defined gradients, a variety of approaches have been proposed. Most of these construct surrogate loss functions via smoothing the optimization problem [18, 25, 9, 26, 23, 15]. Alternatively, Elmachtoub and Grigas [8] propose a closed-form convex surrogate loss with desirable theoretical properties; however, this loss applies only when the optimization problem has a linear objective function. Similarly, Mulamba et al. [20] provide a contrastive learning-based surrogate which does not require differentiation through optimization, but which is also developed specifically for linear objectives. When the predictive model is itself a linear function, Guler et al. [11] propose an approach to search directly for the best model.

Perhaps the most related work to ours is by Berthet et al. [4]. They differentiate through linear optimization problems during training by adding randomized perturbations to smooth the induced loss function. Specifically, they randomly perturb the predictions $\hat{y}$ with random noise $\epsilon$ and solve for $z^*(\hat{y} + \epsilon)$, which can be interpreted as replacing the decision loss with its averaged value in a neighborhood around $\hat{y}$, where the averaging smooths the function and ensures differentiability. There are two key differences between our approach and theirs. First, they apply random perturbations to optimization *in the training loop* in order to produce a smoother surrogate loss. By contrast, we use random perturbations to learn a loss function *prior to training*; during training optimization is removed entirely. This allows us to use the same random samples to inform each training iteration instead of drawing new samples per iteration. Second, their approach applies only to linear objective functions while ours applies to arbitrary optimization problems.

# 4   Locally Optimized Decision Losses (LODL)

In this paper, we do away with the need for custom task-specific relaxations of the optimization problem $z^*(\hat{y})$ by instead translating task-specific information from the decision loss $DL$ into a loss function $LODL_\phi(\hat{y}, y)$ that (i) approximates the behavior of $DL$, *and* (ii) is convex by construction.

Our broad strategy to do this is to learn the function $LODL_\phi \approx DL$ using supervised machine learning. Specifically, we proceed in 3 steps:

1. **We simplify the learning problem in two ways (Section 4.1)**. First, we learn a separate LODL for every $((x, y))$ pair in the dataset to make our learning problem easier. Second, we note that there's a chicken-and-egg problem associated with learning LODLs—to train the LODL we need inputs of the form $(\hat{y}, y)$, but to produce $\hat{y}$ we need a predictive model trained on said LODL. To resolve this, we make the assumption that our predictive model $M_\theta$ will get us sufficiently close to the true labels $y$. This means:

$$LODL_\phi(\hat{y}, y) = [LODL_{\phi_1}(\hat{y}_1), \ldots, LODL_{\phi_N}(\hat{y}_N)], \quad \text{and} \quad \hat{y}_i \approx y_i \pm \epsilon$$

2. Given these simplifications, **we propose convex-by-construction parametric forms for $LODL_{\phi_i}$ (Section 4.2)**. We subtract a constant $DL(y_i, y_i)$ from the target to ensure that the function to be learned has a minima at $\hat{y}_i = y_i$ and that the result can be modeled well by a convex function:

$$LODL_{\phi_i}(\hat{y}_i) \approx DL(\hat{y}_i, y_i) - DL(y_i, y_i) \implies DL(\hat{y}_i, y_i) \approx LODL_{\phi_i}(\hat{y}_i) + \overbrace{DL(y_i, y_i)}^{\text{constant}}$$

3. **We propose different strategies for sampling $\hat{y}$ (Section 4.3)** and also describe the equation used to learn the optimal LODL parameters $\phi^*$ (Equation 2).

## 4.1   Local Loss Functions

We introduce a *separate* set of parameters $\phi_n$ for each $[(x_1, y_1), \ldots, (x_n, y_n), \ldots, (x_N, y_N)]$ in the training set. We take this step because learning a *global* approximation to $DL(\hat{y}, y)$ (for arbitrary $\hat{y}$) is hard; it requires learning a closed-form approximation to the general optimization problem $z^*(\hat{y})$ which may not always exist. Introducing separate parameters per-instance gives two key advantages.

1. **Learning for a specific $y_n$:** Instead of learning a $\mathbb{R}^{\dim(\boldsymbol{y})+\dim(\hat{\boldsymbol{y}})}$ function $LODL_\phi(\hat{\boldsymbol{y}}, \boldsymbol{y})$ to imitate $DL(\hat{\boldsymbol{y}}, \boldsymbol{y})$, we instead learn $N$ different $\mathbb{R}^{\dim(\hat{\boldsymbol{y}})}$ functions $LODL_{\phi_n}(\hat{\boldsymbol{y}})$ that imitate $DL(\hat{\boldsymbol{y}}, \boldsymbol{y}_n)$ for each $n \in N$. Doing this significantly reduces the dimensionality of the learning problem—this is especially relevant when $N$ is not very large in comparison to $\dim(\boldsymbol{y})$ (as in our experiments). It also circumvents the need to enforce invariance properties on the global loss. For example, many optimization problems are invariant to permutations of the ordering of dimensions in $\boldsymbol{y}$, making it difficult to measure the quality of $\hat{\boldsymbol{y}}$'s across different instances. In a local loss, the ordering of the dimensions is fixed and so this issue is no longer relevant.

2. **Learning for only $\hat{\boldsymbol{y}}_n \approx \boldsymbol{y}_n$:** In addition to the simplification above, we don't try to learn a faithful approximation $\forall \hat{\boldsymbol{y}} \in \mathbb{R}^{\dim(\boldsymbol{y})}$—we instead limit ourselves to learning an approximation of $DL(\hat{\boldsymbol{y}}, \boldsymbol{y}_n)$ only when $\hat{\boldsymbol{y}}$ *is in the neighborhood of* $\boldsymbol{y}_n$. We assume that our predictive model will always get us in the neighborhood of the true labels, and the utility of LODL is in helping distinguish between these points.

The combination of these two choices makes the learned $LODL_{\phi_n}$ a "local" surrogate for $DL$, rather than a global one.

## 4.2 Representing the $LODL$

The key design choice in instantiating our framework is choice of the parametric family used to represent the LODL. Optimization problems can induce a complex loss landscape which is not easily summarized in a closed-form function with concise parameterization. Accordingly, we design a set of families which capture phenomena particularly important for common families of predict-then-optimize problems.

**Parametric families for the local losses** Having made the decision to allow separate parameters for each training instance, the second design choice is how to represent each local loss, i.e., the specific parametric family to use. Our choice must be sufficiently expressive to capture the local dynamics of the optimization problem while remaining sufficiently efficient to be replicated across the $N$ training instances. We propose that the structure of the loss function should capture the underlying rationale for why decision-focused learning provides an advantage over 2-stage in the first place; this is the key behavior which will underpin improved decision quality. We identify three key phenomena which motivate the design of the family of loss functions:

1. **Relative importance of different dimensions:** Typically, the different dimensions of a prediction problem are given equal weight, e.g., the MSE weights errors in each coordinate of $\boldsymbol{y}$ equally. However, there may be some dimensions along which $DL$ is more sensitive to local perturbations. For example, a knapsack problem may be especially sensitive to errors in the value of items which are on the cusp of being chosen. In such cases, DFL can capture the relative importance of accurately predicting different dimensions.

2. **Cost of correlation:** Given the possibly large dimensionality of $\boldsymbol{y}$, in practice, the predictive model $M_{\boldsymbol{\theta}}$ does not typically predict $\boldsymbol{y}$ directly. Instead, the the structure of the optimization is exploited to make multiple predictions $[\hat{y}_1, \ldots, \hat{y}_K]$ that are then combined to create $\hat{\boldsymbol{y}}$. For example, in a knapsack problem, we might train a model which separately predicts the value of each item (i.e., predicts each entry of $\hat{\boldsymbol{y}}$ separately) using features specific to that item, instead of jointly predicting the entire set of values using the features of all of the items. However, Cameron et al. [5] show that ignoring the correlation between different sub-$\boldsymbol{y}$ scale predictions (as in 2-stage) can result in poor optimization performance, and that DFL can improve by propagating information about the interactions between entries of $\boldsymbol{y}$ to the predictive model.

3. **Directionality of predictions:** In the $\arg\min(\hat{y}_1, \hat{y}_2)$ example from the introduction, the prediction $\hat{y}_1 = 2, \hat{y}_2 = 1$ produces the same decision as $\hat{y}_1 = 2000, \hat{y}_2 = 1$. On the other hand, the prediction $\hat{y}_1 = 0.5, \hat{y}_2 = 1$ leads to a different decision. As a result, over-predicting and under-predicting often have different associated costs for some optimization problems. Predictive models trained with DFL can take into account this behavior while those trained by typical symmetric 2-Stage losses like MSE cannot.

Given these insights, we propose three corresponding families of loss functions for $LODL_\phi$, each of which is convex by design and has a global minima at the true label $\boldsymbol{y}_n$, a desirable property because $DL$ also has its minima at $\hat{\boldsymbol{y}} = \boldsymbol{y}$.

1. **WeightedMSE:** To take into account the relative importance of different dimensions, we propose a weighted version of MSE:

$$\text{WeightedMSE}(\hat{\boldsymbol{y}}) = \sum_{l=1}^{\dim(\boldsymbol{y})} w_l \cdot (\hat{y}_l - y_l)^2,$$

where 'weights' $w_l$ are the parameters of the LODL, i.e., $\boldsymbol{\phi} = \boldsymbol{w} \in \mathbb{R}_+^{\dim(\boldsymbol{y})}$ (for convexity).

2. **Quadratic:** To take into account the effect of correlation of different dimensions on each other, we propose learning a quadratic function that has terms of the form $(\hat{y}_i - y_i)(\hat{y}_j - y_j)$:

$$\text{Quadratic}(\hat{\boldsymbol{y}}) = (\hat{\boldsymbol{y}} - \boldsymbol{y})^T H (\hat{\boldsymbol{y}} - \boldsymbol{y}),$$

where $\boldsymbol{\phi} = H$ is a learned low-rank symmetric Positive semidefinite (PSD) matrix. This family of functions is convex as long as $H$ is PSD, which we enforce by parameterizing $H = L^T L$, where $L$ is a low-rank triangular matrix $L$ of dimension $\dim(\boldsymbol{y}) \times k$ and $k$ is the desired rank.

This loss function family has an appealing interpretation because learning LODL is similar to estimating the partial derivative of $DL$ with respect to its first input $\hat{\boldsymbol{y}}_n$. Specifically, consider the first three terms of the Taylor expansion of $DL$ with respect to $\hat{\boldsymbol{y}}_n$ at $(\boldsymbol{y}_n, \boldsymbol{y}_n)$:

$$DL(\overbrace{\boldsymbol{y}_n + \boldsymbol{\epsilon}}^{\hat{\boldsymbol{y}}_n}, \boldsymbol{y}_n) = \overbrace{DL(\boldsymbol{y}_n, \boldsymbol{y}_n)}^{\text{constant}} + \overbrace{\nabla_{\hat{\boldsymbol{y}}_n} DL(\boldsymbol{y}_n, \boldsymbol{y}_n)}^{0 \leftarrow (\boldsymbol{y}_n, \boldsymbol{y}_n) \text{ is a minima}} \boldsymbol{\epsilon} + \boldsymbol{\epsilon}^T \overbrace{\nabla_{\hat{\boldsymbol{y}}_n}^2 DL(\boldsymbol{y}_n, \boldsymbol{y}_n)}^{\text{Hessian } H} \boldsymbol{\epsilon} + \dots$$

$$\approx DL(\boldsymbol{y}_n, \boldsymbol{y}_n) + (\hat{\boldsymbol{y}}_n - \boldsymbol{y}_n)^T H (\hat{\boldsymbol{y}}_n - \boldsymbol{y}_n)$$

Quadratic $LODL_\phi$ can be seen as a $2^{nd}$-order Taylor-series approximation of $DL$ at $(\boldsymbol{y}_n, \boldsymbol{y}_n)$ where the learned $H$ approximates the Hessian of $DL$. Note that WeightedMSE is a special case of this Quadratic loss when $H = diag(\boldsymbol{w})$.

3. **DirectedWeightedMSE and DirectedQuadratic:** To take into account the fact that overpredicting and underpredicting can have different consequences, we propose modifications to the two loss function families above. For WeightedMSE, we redefine the weight vector $\boldsymbol{w}$ as below, and learn both $\boldsymbol{w}_+$ and $\boldsymbol{w}_-$. Similarly for Quadratic, we define 4 copies of the parameter $L$ based on the directionality of the predictions.

$$w_l = \begin{cases} w_+, & \text{if } \hat{y}_i - y_i \geq 0 \\ w_-, & \text{otherwise} \end{cases} \qquad L_{ij} = \begin{cases} L_{ij}^{++}, & \text{if } \hat{y}_i - y_i \geq 0 \text{ and } \hat{y}_j - y_j \geq 0 \\ L_{ij}^{+-}, & \text{if } \hat{y}_i - y_i \geq 0 \text{ and } \hat{y}_j - y_j < 0 \\ L_{ij}^{-+}, & \text{if } \hat{y}_i - y_i < 0 \text{ and } \hat{y}_j - y_j \geq 0 \\ L_{ij}^{--}, & \text{otherwise} \end{cases}$$

### 4.3 Learning $LODL_\phi$

Given families proposed in Section 4.1, our goal is to learn some $\phi_n^*$ for every $n \in N$ such that $LODL_{\phi_n}(\hat{\boldsymbol{y}}_n) \approx DL(\hat{\boldsymbol{y}}_n, \boldsymbol{y}_n)$ for $\hat{\boldsymbol{y}}_n$ "close" to $\boldsymbol{y}_n$. We propose a supervised approach to learning $\phi_n^*$ which proceeds in two steps (Figure 1): (1) we build a dataset mapping $\hat{\boldsymbol{y}}_n \to DL(\hat{\boldsymbol{y}}_n, \boldsymbol{y}_n)$ in the region of $\boldsymbol{y}_n$, and then (2) we use this dataset to estimate $\phi_n^*$ by minimizing the mean squared error to the true decision loss:

$$\phi_n^* = \arg\min_{\phi_n} \frac{1}{K} \sum_{k=1}^K (LODL_{\phi_n}(\boldsymbol{y}_n^k) - DL(\boldsymbol{y}_n^k, \boldsymbol{y}_n))^2 \qquad (2)$$

This framework has the key advantage of reducing the design of good surrogate tasks (a complex problem requiring in-depth knowledge of each optimization problem) to supervised learning (for which many methods are available). Indeed, future advances, e.g. in representation learning, can simply be plugged into our framework.

The major remaining step is to specify the construction of the dataset for supervised learning of $\phi_n^*$. We propose to construct this dataset by sampling a set of $K$ points $[\boldsymbol{y}_n^1, \dots, \boldsymbol{y}_n^K]$ in the vicinity of $\boldsymbol{y}$ and calculate $DL(\boldsymbol{y}_n^k, \boldsymbol{y}_n)$ for each. In this paper, we consider three sampling strategies:

1. **All-Perturbed:** Add zero-mean Gaussian noise to the true label $\boldsymbol{y}_n$:

$$\boldsymbol{y}_n^i = \boldsymbol{y}_n + \boldsymbol{\epsilon}^k = \boldsymbol{y}_n + \alpha \cdot \mathcal{N}(0, I),$$

where $\alpha$ is a normalization factor and $I$ is a $dim(\boldsymbol{y}) \times dim(\boldsymbol{y})$ identity matrix.

2. **1-Perturbed or 2-Perturbed:** Estimating the behavior of $DL(\boldsymbol{y}_n + \epsilon^i, \boldsymbol{y}_n)$ for small $\epsilon$ can alternatively be interpreted as estimating $(\frac{\delta}{\delta \hat{\boldsymbol{y}}_n})^{dim(\boldsymbol{y}_n)} DL(\hat{\boldsymbol{y}}_n, \boldsymbol{y}_n)$ (i.e., the $dim(\boldsymbol{y}_n)^{th}$ partial derivative of $DL$ w.r.t. its first input $\hat{\boldsymbol{y}}_n$ at $(\boldsymbol{y}_n, \boldsymbol{y}_n)$ because all the dimensions of $\hat{\boldsymbol{y}}_n$ are being varied simultaneously. While computing $(\frac{\delta}{\delta \hat{\boldsymbol{y}}_n})^{dim(\boldsymbol{y}_n)}$ is computationally challenging, it can be estimated using the simpler 1$^{st}$ or 2$^{nd}$ partial derivatives. This corresponds to perturbing only one or two dimensions at a time.

## 4.4 Time Complexity of Learning LODLs

The amount of time taken by each of the methods using gradient descent is:

- **2-Stage** = $\Theta(T \cdot N \cdot T_M)$, where $T_M$ is the amount of time taken to run one forward and backwards pass through the model $M_\theta$ for one optimization instance, $N$ is the number of optimization instances, and $T$ is the number of time-steps $M_\theta$ is trained for.

- **DFL** = $\Theta(T \cdot N \cdot (T_M + T_O + T'_O))$, where $T_O$ is the time taken to solve the forward pass of one optimization instance and $T'_O$ is the time taken to compute the backward pass.

- **LODL** = $\Theta(K \cdot N \cdot T_O + N \cdot (T \cdot K \cdot T_{LODL}) + T \cdot N \cdot T_M)$, where $K$ is the number of samples needed to train the LODL, and $T_{LODL}$ is the amount of time taken to run one forward and backwards pass through the LODL. The three terms correspond to (i) generating samples, (ii) training $N$ LODLs, and (iii) training $M_\theta$ using the trained LODLs.

In practice, we find that $(T'_O > T_O) >> (T_M > T_{LODL})$. As a result, the difference in complexity of DFL and LODL is roughly $\Theta(T \cdot N \cdot T_O)$ vs. $\Theta(K \cdot N \cdot T_O + N \cdot T \cdot K \cdot T_{LODL})$. Further, the calculation above assumes that LODLs are trained in the same way as $M_\theta$. However, in practice, they can often be learned much faster, sometimes even in closed form (e.g., WeightedMSE and DirectedWeightedMSE), leading to an effective runtime of $\Theta(K \cdot N \cdot T_O + N \cdot K \cdot T_{LODL}) \approx \Theta(K \cdot N \cdot T_O)$. *Then, the difference between DFL and LODL boils down to $T$ vs. $K$, i.e., the number of time-steps needed to train $M_\theta$ vs. the number of samples needed to train the LODL.*

While our approach *can* be more computationally expensive, it typically isn't for two reasons:

- **Amortization:** We need only sample candidate predictions once, to then train *any number* of LODLs (e.g., WeightedMSE, DirectedQuadratic) without ever having to call an optimization oracle. Once the LODLs have been learned, you can train *any number* of predictive models $M_\theta$ based on said LODLs—in contrast to DFL, which requires calling the oracle to train *each* model. DFL is thus more expensive when training a large number of models (e.g., for hyperparameter/architecture search, trading-off performance vs. inference time vs. interpretability, etc.). *In the future, we imagine that datasets could be shipped with not only features and labels, but also LODLs associated with downstream tasks!*

- **Parallelizability:** The sample generation process for LODL is completely parallelizable, resulting in an $\Omega(T_O)$ lower-bound wall-clock complexity for our approach. In contrast, the calls to the optimization oracle in DFL are interleaved with the training of $M_\theta$ and, as a result, cannot be parallelized with respect to $T$, resulting in an $\Omega(T \cdot T_O)$ wall-clock complexity.

We demonstrate this empirically in Section 5.3.

## 5 Experiments

To validate the efficacy of our approach, we run experiments on three resource allocation tasks from the literature. We use the term *decision quality $DQ$* (higher is better) instead of $DL$ because these are all maximization problems.

**Linear Model**     This domain involves learning a linear model when the underlying mapping between features and predictions is cubic. Such problems are common in the explainable AI literature [21, 10, 12] where predictive models must be interpretable.

- *Predict:* Given a feature $x_n \sim U[0, 1]$, use a linear model to predict the utility $\hat{y}$ of resource $n$, where the true utility is $y_n = 10x_n^3 - 6.5x_n$. Combining predictions yields $\hat{\boldsymbol{y}} = [\hat{y}_1, \ldots, \hat{y}_N]$.

- *Optimize:* Choose the $B = 1$ out of $N = 50$ resources with highest utility: $\boldsymbol{z}^*(\hat{\boldsymbol{y}}) = \arg\text{topk}(\hat{\boldsymbol{y}})$

Table 1: The decision quality achieved by each approach—**higher is better**. DirectedQuadratic consistently performs well.

| Loss Function | Normalized $DQ$ On Test Data | | |
| --- | --- | --- | --- |
| | Linear Model | Web Advertising | Portfolio Optimization |
| Random | 0 | 0 | 0 |
| Optimal | 1 | 1 | 1 |
| 2-Stage (MSE) | -0.95 ± 0.00 | 0.48 ± 0.15 | 0.32 ± 0.02 |
| DFL | 0.83 ± 0.38 | 0.85 ± 0.10 | **0.35 ± 0.02** |
| NN | **0.96 ± 0.00** | 0.81 ± 0.14 | -0.11 ± 0.08 |
| WeightedMSE | -0.93 ± 0.06 | 0.58 ± 0.15 | 0.31 ± 0.02 |
| DirectedWeightedMSE | **0.96 ± 0.00** | 0.53 ± 0.14 | 0.32 ± 0.02 |
| Quadratic | -0.75 ± 0.38 | **0.93 ± 0.04** | 0.27 ± 0.02 |
| DirectedQuadratic | **0.96 ± 0.00** | 0.91 ± 0.04 | 0.33 ± 0.01 |

- *Surrogate:* Because the $\arg\max$ operation is piecewise constant, DFL requires a surrogate—we use the soft Top-K proposed by Xie et al. [27]. Although this surrogate is convex in the decision variables, it is *not* convex in the predictions.

**Web Advertising**    This is a submodular optimization task taken from Wilder et al. [25]. The aim is to determine on which websites to advertise given features about different websites.

- *Predict:* Given features $x_m$ associated with some website $m$, predict the click-through rates (CTRs) for a fixed set of $N = 10$ users on $M = 5$ websites $\hat{y}_m = [\hat{y}_{m,1}, \ldots, \hat{y}_{m,N}]$. To obtain the features $x_m$ for each website $m$, true CTRs $y_m$ from the Yahoo! Webscope Dataset [28] are scrambled by multiplying with a random $N \times N$ matrix $A$, i.e., $x_m = Ay_m$.
- *Optimize:* Given the matrix of CTRs, determine on which $B = 2$ (budget) websites to advertise such that the expected number of users that click on the ad *at least once* is maximized, i.e., $z^*(\hat{y}) = \arg\max_z \sum_{j=0}^{N}(1 - \prod_{i=0}^{M}(1 - z_i \cdot \hat{y}_{ij}))$, where all the $z_i \in \{0, 1\}$.
- *Surrogate:* Instead of requiring that $z_i \in \{0, 1\}$, the multi-linear relaxation from Wilder et al. [25] allows fractional values. The $DL$ induced by the relaxation is *non-convex*.

**Portfolio Optimization**    This is a Quadratic Programming domain [7, 24] in which the aim is to choose a distribution over $N$ stocks that maximizes the expected profit minus a quadratic risk penalty. We choose this domain as a stress test—it is highly favorable for DFL because the optimization problem naturally provides informative gradients and thus requires no surrogate.

- *Predict:* Given historical data $x_n$ about stock $n$, predict the future stock price $y_n$. We use historical data from 2004 to 2017 for a set of $N = 50$ stocks from the QuandlWIKI dataset [22].
- *Optimize:* Given a historical correlation matrix $Q$ between pairs of stocks, choose a distribution $z$ over stocks that maximizes $z^T y - \lambda \cdot \hat{y}^T Q \hat{y}$, where $\lambda = 0.1$ is the risk aversion constant.

More experimental setup details are provided in Appendix A.

## 5.1   Results

We train either a linear model (for the Linear Model domain) or a 2-layer fully-connected neural network with 500 hidden units (for the other domains) using LODLs and compare it to:

1. **Random**: The predictions are sampled uniformly from $[0, 1]^{dim(y)}$.
2. **Optimal**: The predictions are equal to the true labels $y$.
3. **2-Stage**: The model is trained on the standard MSE loss ($\frac{1}{N} \sum_{n=1}^{N} ||\hat{y}_n - y_n||^2$).
4. **DFL**: Decision-focused learning using the specified surrogate.
5. **NN:** To determine how important convexity is for LODL we define $LODL_\phi = NN_\phi(\hat{y})$ in which $NN_\phi$ is a 4-layer fully-connected Neural Network (NN) with 100 hidden units.

Table 1 shows the main results. We find that, in all domains, training predictive models with LODL outperforms training them with a task-independent 2-stage loss. Surprisingly, it also outperforms DFL, which has the benefit of handcrafted surrogates, in two of the three. This is strong evidence in favor of our hypothesis that we can automate away the need for handcrafting surrogates. We first analyze the results in terms of the domains:

Figure 3: Figure (a) shows that the time taken to train a single model $M_\theta$ reduces with the number of cores used. Figure (b) shows that this time is *further* reduced when the cost of learning LODLs is amortized across different predictive models $M_\theta$.

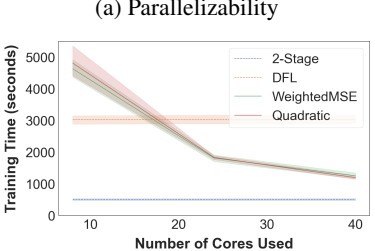
(a) Parallelizability

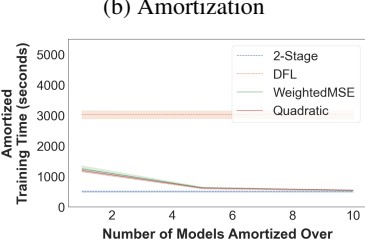
(b) Amortization

1. **Linear Model:** In this domain, the directionality of predictions is very important. As we describe in Section 2.1, predicting values higher than the true utilities for the $B$ resources with highest utility, does not change the decision. However, if their predicted utility is lower than that of the $B - 1^{\text{th}}$ best resource, the decision quality is affected. As a result, we see that the "Directed" methods perform significantly better than their competition.

2. **Web Advertising**: In this domain, Wilder et al. [25] suggest that the decision quality is linked to being able to predict the quantity $\sum_j \hat{y}_{ij}$ (the sum of CTRs across all users $j$ for a given website $i$). However, because the input features for every $\hat{y}_{ij}$ are the same $x_i$, the errors can be correlated and so the sum can be biased. As a result, the ability to penalize the correlations between two predictions $\hat{y}_{ij}$ and $\hat{y}_{jk}$ is important to being able to perform well on this task—which results in the Quadratic methods outperforming the others.

3. **Portfolio Optimization**: Given that this stress-test domain was built to be favorable to $DFL$, it outperforms all other approaches with statistical significance. While the directed LODL methods do not outperform DFL, they nonetheless significantly outperform 2-stage at $p < 0.05$.

We now analyze the results in terms of the methods:

1. **Our DirectedQuadratic $LODL$ consistently does well:** In addition to consistently high expected values (always better than 2-stage), the associated variance is lower as well.

2. **Lack of convexity can cause inconsistent results**: While NN does well in the first two domains, it fails catastrophically in the Portfolio Optimization domain.

3. **DFL has a large variance in performance**: In both the "Web Advertising" and "Linear Model" domain, DFL has higher variation than the best performing LODLs. We posit that this is also because of the lack of convexity of the surrogates that DFL uses in these two domains.

## 5.2 Ablations

We study the impact of the sampling strategy and number of samples on the performance of the LODL methods in the Web Advertising domain in Table 3. We find:

1. **Sampling strategy (Table 3a):** *The best sampling strategy is loss family-specific.* Specifically, NN and DirectedWeightedMSE perform best with the "2-Perturbed" strategy, while the remaining LODLs perform best with the "All-Perturbed" strategy.

2. **Number of samples (Table 3b):** All models perform better with more samples. In addition, the variance reduces as the number of samples increases (especially for Quadratic), suggesting that better approximations of $DL$ lead to more consistent outcomes.

## 5.3 Computational Cost of Learning with LODLs

We measure the time taken to learn predictive models $M_\theta$ with LODLs in the Web Advertisement domain. We train each LODL for 100 gradient descent steps using 5000 samples and train the predictive model for 500 steps (the same as the setup as Table 1). We find:

1. **Learning LODLs is parallelizable:** Figure 3a shows that the cost of training a predictive model using LODLs decreases near-linearly in the number of cores used. With more than 20 cores, *training with LODLs can be cheaper than training with DFL for this domain.*

2. **If LODLs can be reused, their (already low) overhead quickly diminishes**: From Figure 3b we see that even for a modest amount of amortization (over 5-10 predictive models), the training time using LODLs converges to that of two-stage (shown using parallelization with 40 cores).

### 5.4 Correlation between the 'quality' of LODL and decision quality

Recall that LODL losses are fit using a Gaussian sampling strategy centered around the true labels. It is natural to ask how well this proxy loss correlates with the decision quality on test data. We do this by measuring the mean absolute error (MAE) of LODL relative to the ground truth decision loss for points in the *Gaussian neighborhood* around the true labels.

This Gaussian neighborhood is only an approximation of the true distribution of interest—the distribution of predictions generated by the predictive model that is trained using the LODL loss. We can measure the MAE on this distribution, which we call the *empirical neighborhood*.

Table 2: A table showing the relationship between the quality of the learned loss for different classes of LODLs, and the performance of a model trained on said loss. The Empirical Neighborhood MAE is linearly correlated with $DQ$ while the Gaussian Neighborhood MAE is not.

| Approach | MAE in Gaussian Neighborhood (x $10^{-2}$) | MAE in Empirical Neighborhood (x $10^{-2}$) | Normalized DQ on Test Data | |
|---|---|---|---|---|
| NN | $0.94 \pm 0.06$ | $2.22 \pm 1.73$ | $0.80 \pm 0.16$ | 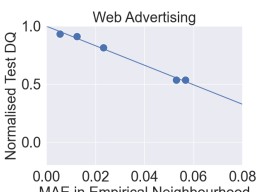 |
| WeightedMSE | $1.04 \pm 0.00$ | $4.48 \pm 1.71$ | $0.58 \pm 0.15$ | |
| DirectedWeightedMSE | $\mathbf{0.92 \pm 0.00}$ | $5.58 \pm 1.64$ | $0.50 \pm 0.13$ | |
| Quadratic | $0.96 \pm 0.00$ | $\mathbf{0.86 \pm 0.52}$ | $\mathbf{0.92 \pm 0.05}$ | |
| DirectedQuadratic | $1.06 \pm 0.00$ | $1.91 \pm 0.79$ | $0.85 \pm 0.08$ | |

Table 2 shows the results for the Budget Allocation domain, while the remaining graphs are in Appendix B.3. All methods are able to approximate the $DL$ comparably well in the Gaussian neighborhood, but this does not correlate well with decision quality. In contrast, the error on the empirical neighborhood is tightly linearly correlated with decision quality. Furthermore, if we extrapolate the line of best fit to where the MAE is 0, i.e., when there is no discrepancy LODL and $DL$, we find that the trend predicts the normalized $DQ$ would be 1.

## 6 Discussion and Conclusion

Our work proposes a conceptual shift from hand-crafting surrogate losses for decision problems to automatically learning them, and demonstrates experimentally that the LODL paradigm enables us to learn high-quality models without such manual effort. Nevertheless, our current instantiation of this framework has limitations which are areas for future work.

We considered LODLs that additively decompose across the dimensions of $\boldsymbol{y}$, allowing us to isolate the effects of fit to $DL$ from the generalization performance across the dimensions of $\boldsymbol{y}$. Future work may learn models that generalize across dimensions, allowing for even greater scalability.

We demonstrate that the fit of a LODL to the empirical neighborhood around the ground truth label is highly correlated with the eventual decision quality. While the Gaussian neighborhood method does yield models that perform well, it does not correlate well with the decision quality across LODL parameterizations. It would be valuable to study the empirical neighborhood to better understand the reasons for this discrepancy and potentially develop LODLs with even stronger performance.

In summary, LODL provides an alternate framework for machine learning which informs decision making, opening up new avenues towards models which are both high-performing and easily trained.

## Acknowledgments and Disclosure of Funding

Research was sponsored by the ARO and was accomplished under Grant Number: W911NF-18-1-0208. Wilder was supported by the Schmidt Science Fellows program.

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
