# A  Extended Experimental Setup

We provide an extended version of the Experimental Setup from Section 5 below.

**Linear Model**   This domain involves learning a linear model when the underlying mapping between features and predictions is cubic. Concretely, the aim is to choose the top $B = 1$ out of $N = 50$ resources using a linear model. The fact that the features can be seen as 1-dimensional allows us to visualize the learned models (as seen in Figure 4).

Predict: Given a feature $x_n \sim U[0, 1]$, use a linear model to predict the utility $\hat{y}$ of choosing resource $n$, where the true utility is given by $y_n = 10x_n^3 - 6.5x_n$. Combining predictions yields $\hat{\boldsymbol{y}} = [\hat{y}_1, \ldots, \hat{y}_N]$. There are 200 $(\boldsymbol{x}, \boldsymbol{y})$ pairs in each of the training and validation sets, and 400 $(\boldsymbol{x}, \boldsymbol{y})$ pairs in the test set.

Optimize: Given these predictions, choose the $B = 1$ (budget) resources with the highest utility:

$$\boldsymbol{z}^*(\hat{\boldsymbol{y}}) = \arg \operatorname{topk}(\hat{\boldsymbol{y}}).$$

Surrogate: Because the $\arg \max$ operation is piecewise constant, DFL requires a surrogate—we use the soft Top-K proposed by Xie et al. [27] that reframes the Top-K problem with entropy regularization as an optimal transport problem. Note that this surrogate is *not* convex in the predictions.

Intuition: With limited model capacity, you cannot model *all* the data accurately. Better performance can be achieved by modeling the aspects of the data that are most relevant to decision-making—in this case, the behavior of the top $2\%$ of resources. Such problems are common in the explainable AI literature [21, 10, 12] where predictive models must be interpretable and so model capacity is limited.

**Web Advertising**   This is a submodular optimization task taken from Wilder et al. [25]. The aim is to determine on which $B = 2$ websites to advertise given features about $M = 5$ different websites. The predictive model being used is a 2-layer feedforward neural network with an intermediate dimension of 500 and ReLU activations.

Predict: Given features $\boldsymbol{x}_m$ associated with some website $m$, predict the clickthrough rates (CTRs) for a fixed set of $N = 10$ users $\hat{\boldsymbol{y}}_m = [\hat{y}_{m,1}, \ldots, \hat{y}_{m,N}]$. These CTR predictions for each of the $M = 5$ websites are stitched together to create an $M \times N$ matrix of CTRs $\hat{\boldsymbol{y}}$. The task is based on the Yahoo! Webscope Dataset [28] which contains multiple CTR matrices. We randomly sample $M$ rows and $N$ columns from each matrix and then split the dataset such that the training, validation and test sets have 80, 20 and 500 matrices each. To generate the features $\boldsymbol{x}_m$ for some website $m$, the true CTRs $\boldsymbol{y}_m$ for the website are scrambled by multiplying with a random $N \times N$ matrix $A$, i.e., $\boldsymbol{x}_m = A\boldsymbol{y}_m$.

Optimize: Given this matrix of CTRs, determine on which $B = 2$ (budget) websites to advertise such that the expected number of users that click on the advertisement *at least once* is maximized:

$$\boldsymbol{z}^*(\hat{\boldsymbol{y}}) = \arg \max_{\boldsymbol{z}} \ \frac{1}{N} \sum_{j=0}^{N} (1 - \prod_{i=0}^{M} (1 - z_i \cdot \hat{y}_{ij}))$$

$$s.t. \ \sum_{i=0}^{M} z_i \leq B \quad \text{and} \quad z_i \in \{0, 1\}, \text{ for } i \in \{1, \ldots, M\}$$

Surrogate: Instead of requiring that $z_i \in \{0, 1\}$, the multi-linear relaxation suggested in Wilder et al. [25] allows fractional values. However, while this relaxation may allow for non-zero gradients, the induced $DL$ is *non-convex* because the term $\prod_{i=0}^{M}(1 - z_i \cdot \hat{y}_{ij})$ in the objective is non-convex in the predictions.

Intuition: In practice, the CTR values are so small that you can approximate $\prod_{i=0}^{M}(1 - z_i \cdot \hat{y}_{ij}) \approx 1 - z_i \cdot \sum_{i=0}^{M} \hat{y}_{ij}$ because the product terms are almost zero, i.e., $\hat{y}_{ij} * \hat{y}_{i'j} \approx 0$. As a result, the goal is to accurately predict $\sum_{i=0}^{M} y_{ij}$, the sum of CTRs across all the users for a given website. However, because the input features for every $y_{ij}$ are the same $\boldsymbol{x}_i$, the errors are correlated. As a result, when you add up the values the errors do not cancel out, leading to biased estimates.

**Portfolio Optimization**  This is a Quadratic Programming domain popular in the literature [7, 24] because it requires no relaxation in order to run DFL. The aim is to choose a distribution over $N = 50$ stocks in a Markowitz portfolio optimization setup [17, 19] that maximizes the expected profit minus a quadratic risk penalty. The predictive model being used is a 2-layer feedforward neural network with a 500-dimensional intermediate layer using ReLU activations, followed by an output layer with a 'tanh' activation.

Predict: Given historical data $x_n$ about some stock $n$ at time-step $t$, predict the stock price $y_n$ at time-step $t + 1$. Combining the predictions $\hat{y}_n$ across a consistent set of $N = 50$ stocks together yields $\hat{y} = [\hat{y}_1, \ldots, \hat{y}_N]$. We use historical price and volume data of S&P500 stocks from 2004 to 2017 downloaded from the QuandlWIKI dataset [22] to generate $x$ and $y$. There are 200 $(x, y)$ pairs in each of the training and validation sets, and 400 $(x, y)$ pairs in the test set.

Optimize: Given a historical correlation matrix $Q$ between pairs of stocks, choose a distribution $z$ over stocks such that the future return $z^T y$ is maximized subject to a quadratic risk penalty $\hat{y}^T Q \hat{y}$:

$$z^*(\hat{y}) = \arg\max_{z} \ z^T y - \lambda \cdot z^T Q z$$

$$s.t. \ \sum_{i=0}^{N} z_i \leq 1 \quad \text{and} \quad 0 \leq z_i \leq 1, \text{ for } i \in \{1, \ldots, N\}$$

where $\lambda = 0.1$ is the risk aversion constant. The intuition behind the penalty is that if two stocks have strongly correlated historical prices, the penalty will be higher, forcing you to hedge your bets.

Intuition Along the lines of Cameron et al. [5], DFL is able to take into account the correlations in predictions between the $N$ different stocks, while 2-stage is not.

## Computation Infrastructure

We ran 100 samples for each (method, domain) pair—we used 10 different random seeds to generate the domain, and for each random seed we trained the $LODL$s and the predictive model $M_\theta$ for 10 random intializations. We ran all the experiments in parallel on an internal cluster. Each individual experiment was performed on an Intel Xeon CPU with 64 cores and 128 GB memory.

# B  Detailed Experimental Results

## B.1  Visualizing the Linear Model Domain

Figure 4: Graphs showing the true (blue) and 100 learned (orange) mappings between the features and predictions in the Linear Model domain. 2-stage does badly, DFL typically learns the correct slope but can sometimes randomly fail, and DirectedQuadratic does well.

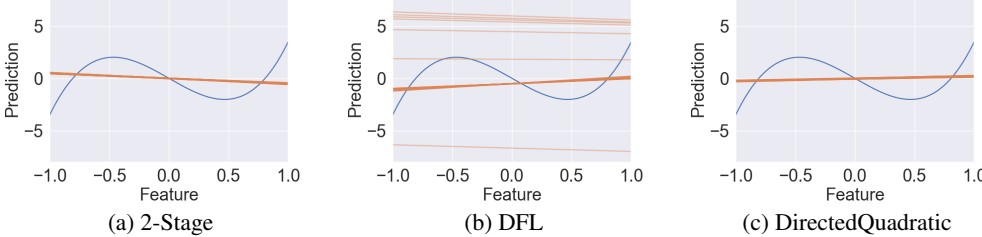

## B.2 Ablations

Table 3: Ablations across (a) different sampling methods and (b) different number of samples in the Web Advertising domain. The best sampling strategy is loss family-specific, while increasing the number of samples uniformly improves the performance.

(a) Across Sampling Strategies

| Approach | Normalized Test $DQ$ | | |
| --- | --- | --- | --- |
| | 1-Perturbed | 2-Perturbed | All-Perturbed |
| NN | $0.86 \pm 0.12$ | $\mathbf{0.89 \pm 0.09}$ | $0.80 \pm 0.16$ |
| WeightedMSE | $0.50 \pm 0.14$ | $0.53 \pm 0.14$ | $\mathbf{0.58 \pm 0.15}$ |
| DirectedWeightedMSE | $0.47 \pm 0.15$ | $\mathbf{0.53 \pm 0.16}$ | $0.50 \pm 0.13$ |
| Quadratic | $0.77 \pm 0.25$ | $0.88 \pm 0.10$ | $\mathbf{0.92 \pm 0.05}$ |
| DirectedQuadratic | $0.77 \pm 0.19$ | $0.84 \pm 0.11$ | $\mathbf{0.85 \pm 0.08}$ |

(b) Across Number of Samples

| Approach | Normalized Test $DQ$ | | |
| --- | --- | --- | --- |
| | 50 samples | 500 samples | 5000 samples |
| NN | $0.81 \pm 0.13$ | $0.80 \pm 0.16$ | $\mathbf{0.81 \pm 0.14}$ |
| WeightedMSE | $0.50 \pm 0.14$ | $0.50 \pm 0.14$ | $\mathbf{0.53 \pm 0.14}$ |
| DirectedWeightedMSE | $0.48 \pm 0.15$ | $0.53 \pm 0.16$ | $\mathbf{0.53 \pm 0.150}$ |
| Quadratic | $0.68 \pm 0.17$ | $0.92 \pm 0.05$ | $\mathbf{0.93 \pm 0.04}$ |
| DirectedQuadratic | $0.59 \pm 0.13$ | $0.85 \pm 0.08$ | $\mathbf{0.91 \pm 0.04}$ |

## B.3 Extending the Results from Section 5.4 to Different Domains

Figure 5 extends the observation that the LODL's goodness of fit in the Empirical Neighborhood linearly correlates to improved 'Decision Quality' to the different domains considered in this paper.

Figure 5: A figure showing the relationship between the quality of the learned LODL and the performance of a model trained on said loss across different domains.

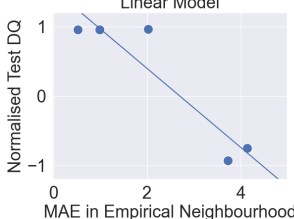 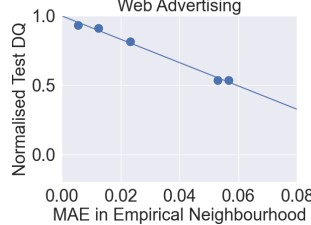 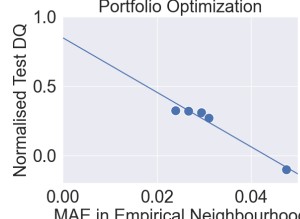