# OpenReview forum: "Decision-Focused Learning without Decision-Making: Learning Locally Optimized Decision Losses"
_NeurIPS.cc/2022/Conference — NeurIPS 2022 Accept_

### Official Review · Reviewer_dkZJ · 2022-07-08

**Rating:** 8
**Confidence:** 4
**Soundness:** 4 excellent
**Presentation:** 4 excellent
**Contribution:** 4 excellent

**Summary:**

This paper considers learning losses for predictive models that
are used in a downstream optimization task.
Section 2 summarizes the basic setup where there is a predictive
model $\hat y=M_\theta(x)$ that creates predictions used to
parameterize an optimization process.
The baselines considered are 1) 2-stage learning, which trains
the predictive model with an intermediate loss, and
2) decision-focused learning, which seeks to optimize a decision loss
with the predictive model, defined with the objective
of the optimization problem.
The locally-optimized decision losses (LODL) proposed in this paper
seek to learn the parameters of surrogate intermediate losses
to match the decision loss.

**Questions:**

I would be very willing to re-evaluate my assessment after a discussion
around the following questions on weaknesses I've listed above.

1. Can you clarify how the LODL loss on line 191 should be interpreted? Did you consider alternatives to this?
2. Can you comment on how the experimental results connect to
   established experimental results in DFL settings?

**Limitations:**

The paper does not clearly discuss limitations in a dedicated section.
While parameterizing and learning an intermediate loss seems appealing,
it seems limited by needing to specify and learn the right
parameterization.

**Strengths And Weaknesses:**

Strengths:
+ The idea of parameterizing surrogate/intermediate losses makes a lot
  of sense in these two-stage settings, and the formulation considered
  here nicely shows the benefits of learning non-Euclidean regression losses.
  I can imagine learned losses to be a crucial,
  impactful, and long-lasting contribution in these settings.
+ The experimental results clearly demonstrate that the LODL is
  able to learn a reasonable surrogate for the tasks considered

Weaknesses:
+ If we take the LODL to be the MSE loss parameterized by weights on
  each dimension, I do not understand the objective on L191:
  my interpretation is that it tries to make the weighted
  MSE loss match the decision loss around the optimal prediction
  by changing the weights.
  Since the MSE loss and decision loss are very different quantities,
  it like this objective will not be possible to optimize.
+ Even though the paper experimentally shows that LODL works, the
  results are difficult to contextualize and compare to related research.
  For example, the web advertising task starting
  at L222 takes one of the settings from
  [Wilder et al., (2019)](https://ojs.aaai.org/index.php/AAAI/article/view/3982),
  but does not provide or present the results in a way that is
  comparable to the results in that paper. The submission would be
  significantly easier to evaluate in comparison to this work
  if it reproduces exactly Table 1 of Wilder et al., (2019) and
  adds additional lines showing how well LODL performs in
  comparison.

Related work:
+ It could be interesting to connect the work to learned losses
  used in meta-learning, such as in
  [Bechtle et al. (2021)](https://arxiv.org/pdf/1906.05374.pdf),
  which takes a much more black-box perspective to learning a
  latent loss function.

---

> ### Author Response · Authors · 2022-08-01
> **Response to Questions**
>
> Thank you for your review and link to related work! Regarding your specific questions:
>
> * **"Why is the equation on L191 a good idea?"**: You're absolutely right that you will never be able to *perfectly* fit the Decision Loss, but you typically don't want to... This is easier to explain with an example:
>     > Consider an $\arg\max$ optimization with 3 'true' parameters (A, B, C), e.g., ${y} = (1, 2, 3)$.
>     > Now, if you predict these parameters perfectly, your 'decision' is ${z} =$ "Pick C", and your 'decision loss' is the true value of parameter C, $DL = 3$.
>     > In fact, any prediction ${\hat{y}}\_{good} = (1 \pm \epsilon\_1, 2 \pm \epsilon\_2, 3 \pm \epsilon\_2)$ for $\epsilon\_1, \epsilon\_2, \epsilon\_2 < 0.5$ will have the exact same decision, and hence exactly the same decision loss. As a result, the decision loss is constant in this region, leading to zero gradients.
>     > While this isn't terrible for these set of predictions, consider the set of predictions ${\hat{y}}\_{bad} = (1 \pm \epsilon\_1, 3 \pm \epsilon\_2, 2 \pm \epsilon\_2)$, where the decision is "Pick B" and the decision loss is 2. If a predictive model makes such a prediction, it cannot improve its predictions because of the zero-gradients. This is why we don't want to fit the decision loss perfectly.
>     >
>     > To understand what WeightedMSE does, consider the two predictions ${\hat{y}}\_{1} = (1, 3.1, 3)$ and ${\hat{y}}\_{2} = (3.1, 2, 3)$. The decision in each of these cases is "Pick A" and "Pick B" respectively, and the corresponding decision losses are 1 and 2. With just these 2 points and the true parameters, WeightedMSE would fit the points $0 \rightarrow 0$ (because 0 error leads to a cost of 0) and  $2.1 \rightarrow 2$ (because adding 2.1 leads to a cost of $3 - 1 = 2$) for parameter A, and $0 \rightarrow 0$ and $1.1 \rightarrow 1$ for parameter B. *As a result, WeightedMSE captures the rough cost of getting each parameter wrong*
>
>     In summary, we try to approximate the Decision Loss by a well-behaved convex function (so it's easy to optimize over) that aims to *roughly* capture the behaviour of the Decision Loss. This is the intuition behind the equation on L191.
>
>     While the approach proposed in the paper you suggested is interesting, learning the full decision loss is as hard as learning the closed-form solution to a (potentially very complex) optimization problem in the Predict-Then-Optimize setting. This is why, in this paper, we focus on designing *Locally-Optimized* Decision Losses for each set of true parameters ${y}$ in the dataset.
>
> * **"How do these results relate to those in the literature?"**: This is a great point and one that we seem to have overlooked in our current description of the experiments. In our paper, 'DFL' corresponds to 'NN2-Decision' from Table 1 in their paper and '2-Stage' corresponds to 'NN2-2Stage'. Given that our aim is just to compare 2-Stage vs. DFL vs. LODL (with similarly structured predictive models), we simplify the structure of Table 1 from Wilder et. al., and re-normalize values so that they are comparable across domains to some extent to create Table 1 in our paper. With regards to our findings, we observe that the results from their paper are sensitive to the choices of different initializations and domain hyperparameters, but observe that (broadly) DFL outperforms 2-Stage. We will make these connections to past work more explicit in the camera-ready.
>
> Also, we briefly discuss the limitations and possible future work in Section 6. While it's true that the *best possible LODL* is task-specific, we believe that:
>
> 1. The phenomena that motivate the choices of the loss function families proposed in the paper (Section 4.1) are fairly general, and thus these LODLs that account for them will be able to outperform 2-Stage methods.
> 2. There is merit to reducing a differentiable optimization problem to a supervised learning problem that more people have expertise in solving (see the common review for more details).

---

> > ### Comment · Reviewer_dkZJ · 2022-08-07
> > **Response on the example in comparison to a regression loss**
> >
> > (I tried to send this immediately after seeing the response but
> > NeurIPS apparently doesn't let reviewers respond until the
> > discussion period officially starts.)
> >
> > Thank you for the clarifications! I am continuing to thoroughly look through all
> > of the other reviewing details and have one clarification question on the example for now. My concern with L191 was also
> > on shaping a supervised/regression loss to the decision loss. In the
> > example you gave, we know the ground-truth
> > parameters $y$, so one option would be to just regress
> > $\mathcal{L}=||M_\theta(x)-y||$. **This regression loss seems easy to optimize and has
> > well-defined gradients.** My interpretation of the LODL loss on L191 is
> > that it parameterizes a regression loss like this and tries to make
> > the regression loss' value match the value of the decision loss in
> > some region around the optimal prediction --- **these seem like very
> > different quantities. This is the part I do not understand how to
> > interpret: L191 seems to be doing more than just smoothing the
> > original decision loss because it tries to make the value of a
> > parameterized regression loss match the decision loss.**

---

> > > ### Author Response · Authors · 2022-08-09
> > > **Clarifications**
> > >
> > > These are great questions! For the first one, *the 2-stage loss (MSE) is exactly $\mathcal{L} = ||M\_\theta(x) - y||\_2\^2$ (up to a normalization factor)!* However, to see why this is bad in the context of the example, consider the 2 predictions $\hat{y}\_{good}=(1, 2, 4.1)$ and $\hat{y}\_{bad} = (1, 3.1, 3)$. Both have the *same error* according to MSE, but have *different values* of the decision loss ($DL(\hat{y}\_{good}) = 3$ and $DL(\hat{y}\_{bad}) = 2$). This is what we mean when we say that the 2-stage and decision losses are 'misaligned'. **As a result, despite the fact that it's easy to optimize and the gradients are well defined, we do not want to minimize $\mathcal{L} = ||M\_\theta(x) - y||$.** Instead, in predict-then-optimize, we ideally want to optimize for the decision loss.
> > >
> > > However, for certain kinds of optimization problems (e.g., linear or discrete optimization) the decision loss cannot be directly optimized for via gradient descent (because of the zero-gradient issue). As a result, you have to come up with some approximation (i.e., *surrogates*) to the decision loss. While past work in DFL *handcrafts* these surrogates, we build task-specific loss functions (LODLs) that satisfy two criteria:
> > >
> > > 1. They are convex-by-construction; we show that having non-convex surrogates leads to bad predictive models $M\_\theta$ (see Section 5.1).
> > > 2. They are close to the Decision Loss (because that is what we want to optimize for).
> > >
> > > **As a result we construct our LODLs such that they are as close to the decision loss while still being convex (and, as a result, easy to optimize over), resulting in the formulation from L191.**
> > >
> > > Does that answer your questions? (Sorry for the delay!)

---

> > > > ### Comment · Reviewer_dkZJ · 2022-08-09
> > > > **Response**
> > > >
> > > > Thank you for the further clarifications! To be honest, I still do not fully understand but will continue thinking about it and will try to further discuss with the other reviewers during the discussion period. If I write the regression-based LODL parameterized by $\phi$ as $\mathcal{L}_\phi=||M_\theta(x)-y||_\phi$, then the loss on L191 of the paper is finding some $\phi$ that optimizes (for a single instance with some $y$ sampled around the true one):
> > > >
> > > > ${\rm argmin}_\phi (||M_\theta(x)-y||_\phi - f(z^\star(y), y))^2$,
> > > >
> > > > where $f(z^\star(y), y)$ is the decision loss. I am sorry to repeat the same point, but I do not see how the value of the regression loss is comparable to the value of the decision loss and why we would want to match the value of a parameterized regression loss to the value of the decision loss. My understanding is that you are saying that I should instead see the LODL as a surrogate to the decision loss rather than a parameterized regression loss.

---

> > > > > ### Author Response · Authors · 2022-08-09
> > > > > **Clarification**
> > > > >
> > > > > Ah! I think I see where we're seeing things differently. The goal in **predict-then-optimize** is to learn some predictive model $M\_{\theta\^\*}$ such that $\theta\^\* = \arg\min_\theta \mathbb{E}\_{(x, y) \sim D} [f(z\^\*(M\_\theta(x)), y)]$ (L77). This expression is what defines what a 'good prediction' is in this setting, i.e., **we want to find a predictive model $M\_{\theta\^\*}$ that has a low decision loss**.
> > > > >
> > > > > Now, it is hard to directly optimize for this because $z\^\*$ is badly behaved, e.g., has zero-gradients. As a result, we come up with some $LODL\_{\phi\^\*}$ that (i) we *can* optimize over, and (ii) if we learn a predictive model $M\_{\theta\^\*_{LODL}}$ such that $\theta\^\*\_{LODL} = \frac{1}{N} \sum\_{(x, y)} LODL\_\phi\^\* (M\_\theta (x), y)$, it approximately optimizes the objective above. **In other words, we want to design a parameterized regression loss (LODL) that mimics the behavior of the decision loss because, if we do, a model $M\_{\theta\^\*_{LODL}}$ that is trained on the LODL can be expected to perform well on the decision loss, and as a result, perform well on the optimization task.**
> > > > >
> > > > > Does that make sense?

---

> > > > > > ### Comment · Reviewer_dkZJ · 2022-08-09
> > > > > > **Response**
> > > > > >
> > > > > > >  In other words, we want to design a parameterized regression loss (LODL) that mimics the behavior of the decision loss
> > > > > >
> > > > > > The misunderstanding I am raising is that the *value* of any of the parameterized regression losses (e.g. 0 at optimality) is going to be extremely different than the *value* of the decision loss. Then why is it reasonable that these values are matched when they are going to be so far off? Does this also create artifacts, for example instances that make the model prioritize instances with larger decision losses because those are farther away than the zero-centered regression loss?
> > > > > >
> > > > > > Does it also make sense to ask if the *curvature* of the regression loss should match the *curvature* of the decision loss around the optimal value?

---

> > > > > > > ### Author Response · Authors · 2022-08-09
> > > > > > > **Response**
> > > > > > >
> > > > > > > I'm not sure that I understand why they would be different? Consider the example from before; we have the points $y = (1, 2, 3)$ with decision loss $3$, $\hat{y}_1 = (1, 3.1, 3)$ with decision loss $2$, and $\hat{y}_2 = (3.1, 2, 3)$ with decision loss $1$ (and we want to maximize the decision loss for this problem). Then, the way that we create input-outputs for the LODL is:
> > > > > > >
> > > > > > > - *Input:* Take the difference between the prediction and the true label. So $\text{input}(y) = (0, 0, 0)$, $\text{input}(\hat{y}_1) = (0, 1.1, 0)$, and $\text{input}(\hat{y}_2) = (2.1, 0, 0)$. This way the true label is always at $(0, 0, 0)$ (for any input).
> > > > > > > - *Output:* Take the difference between the optimal decision loss and the decision loss obtained by the prediction. So $\text{output}(y) = 0$, $\text{output}(\hat{y}_1) = 1$, and $\text{output}(\hat{y}_2) = 2$. This way the optimal value is at 0 for any LODL, and we can also estimate the decision loss using the equation $DL(\hat{y}) \approx DL(y) - LODL(\hat{y})$.
> > > > > > >
> > > > > > > As a result, we enforce (by construction) that $\mathbf{0}$ is the minima of every LODL, that at the minima its value is 0, and this value strictly increases (by convexity). Then, any predictions that are far away from the true labels are penalized strongly, making sure that the predictive model $M_\theta$ makes predictions that are close to the true label.
> > > > > > >
> > > > > > > For values around the true label, the values of the regression loss are similar to the values of the decision loss. For example, by fitting the WeightedMSE to the points $\hat{y}_1$ and $\hat{y}_2$ in the example above, you're guaranteed that $DL(\hat{y}_1) = DL(y) - LODL(\hat{y}_1)$ and $DL(\hat{y}_2) = DL(y) - LODL(\hat{y}_2)$. More generally, by fitting the regression loss using L191, you're guaranteed to get the $\phi^*$ that best matches the $DL$ for the sampled points. This makes the regression loss and decision loss a (somewhat) apples-to-apples comparison.
> > > > > > >
> > > > > > > Now, the choice of loss function family *does* affect how this smoothing is done. For example, using the weighted 2-norm (WeightedMSE) penalizes deviations from the true label quadratically, while a weighted 1-norm would penalize it linearly. However, every supervised learning problem involves finding a model with good "inductive bias", and we think that this challenge is much more approachable to ML practitioners than the challenge of finding a good surrogate optimization problem.
> > > > > > >
> > > > > > > As for curvature, that would be a great thing to measure, but when $z\^\*$ is piecewise constant (as in the example), it's not clear how you would define something like that? Given that we're interested in the cases where optimization problems are "badly behaved" and require surrogates, we do not include such analyses.

---

> > > > > > > > ### Comment · Reviewer_dkZJ · 2022-08-09
> > > > > > > > **Minor last clarification**
> > > > > > > >
> > > > > > > > Thank you for all of the details! I will continue thinking about this throughout the review process. In case you see this in time, one last minor question: how does $DL(\hat{y}) \approx DL(y) - LODL(\hat{y})$ show up in the loss at L191 (or anywhere else in the paper)? It does not appear this connection is made anywhere, and L188 almost seems to contradict this, stating that the goal is for $LODL(\hat{y})\approx DL(\hat{y})$.

---

> > > > > > > > > ### Author Response · Authors · 2022-08-10
> > > > > > > > > **Response**
> > > > > > > > >
> > > > > > > > > It's a good point! We thought that $LODL(\hat{y}) \approx DL(\hat{y})$ was a more intuitive description of what we were trying to do, and ignored the constant term '$DL(y)$' in favor of being 'morally correct'. However, this conversation has been extremely useful; we will change the text to make what we are doing more clear!

---

> > > > > > > > > > ### Comment · Reviewer_dkZJ · 2022-08-10
> > > > > > > > > > **Response**
> > > > > > > > > >
> > > > > > > > > > Ok thank you! That makes so much more sense then. In that case, it seems like the additional $DL(y)$ term cannot be left out of the loss as it's much more of a bias term for the regression rather than a constant that can be ignored. Does this impact any of the experimental results?

---

> > > > > > > > > > > ### Author Response · Authors · 2022-08-10
> > > > > > > > > > > **Response**
> > > > > > > > > > >
> > > > > > > > > > > Nope! We ignored it in the text for notational simplicity, but all the experiments include the term.

---

> > > > > > > > > > > > ### Comment · Reviewer_dkZJ · 2022-08-10
> > > > > > > > > > > > **Raising score from 4 to 8**
> > > > > > > > > > > >
> > > > > > > > > > > > Wow! In that case, I raise my score from a 4 to a 8 and advocate for the paper's acceptance. It's a brilliant idea that will be foundational in the space and will be engaging and thought-provoking at the conference. My original concerns were on 1) how to interpret the loss being optimized and 2) connecting the experimental results to other published results. From this discussion, I trust that the authors will update these significant writing issues in the paper for 1). In their first response in this thread, the authors state that the results are just normalized versions that are directly comparable to Table 1 of Wilder et al. This is an extremely reasonable and grounded experimental setting and in most cases the LODL results in a non-trivial improvement.
> > > > > > > > > > > >
> > > > > > > > > > > > ## On the concerns from other reviewers
> > > > > > > > > > > >
> > > > > > > > > > > > I have read through all of the other reviewing details and do not strongly see a case for rejection from any of the reasons given from the other reviewers, and I am open to a discussion with them, of course. Here is my quick summary of them:
> > > > > > > > > > > >
> > > > > > > > > > > > ### TJiM states:
> > > > > > > > > > > >
> > > > > > > > > > > > > The computational complexity of LODL is high,  LODL is still an approximation-based method to differentiate the optimizer, and analysis/bounds
> > > > > > > > > > > >
> > > > > > > > > > > > I agree with the authors' rebuttal that the experimental results of their method are enough justification for acceptance. Better theoretical understanding is usually helpful, but in this case I do not think it is crucial.
> > > > > > > > > > > >
> > > > > > > > > > > > >  The way to sample and train LODL around each training sample is questionable.
> > > > > > > > > > > >
> > > > > > > > > > > > I agree with this concern and hope the authors will emphasize this is a heuristic part and try to give intuition on what they have found to work and not work.
> > > > > > > > > > > >
> > > > > > > > > > > > ### 4d1n states:
> > > > > > > > > > > >
> > > > > > > > > > > > > I am somewhat skeptical of its practicability considering the challenges in parameter tuning and the computational scalability of the proposed method.
> > > > > > > > > > > >
> > > > > > > > > > > > I think the experimental results justify the method
> > > > > > > > > > > >
> > > > > > > > > > > > ### NGrH states:
> > > > > > > > > > > >
> > > > > > > > > > > > > The design of LODL is interesting, but I have a concern about the complexity of fitting the local loss function
> > > > > > > > > > > >
> > > > > > > > > > > > I think the authors have appropriately acknowledged and addressed thi

---

> > > > > > > > > > > > > ### Comment · Reviewer_dkZJ · 2022-08-10
> > > > > > > > > > > > > **Confirming the submitted code is consistent with the authors response in this thread**
> > > > > > > > > > > > >
> > > > > > > > > > > > > I just also checked the code and verify `losses.py` is consistent with the description in this thread and not as presented in the paper. L170 subtracts the perturbed objectives from the optimal ones:
> > > > > > > > > > > > >
> > > > > > > > > > > > > ```python
> > > > > > > > > > > > > objectives = opt_objective - objectives
> > > > > > > > > > > > > ```
> > > > > > > > > > > > >
> > > > > > > > > > > > > And then L216 computes the regression onto training instances of these:
> > > > > > > > > > > > >
> > > > > > > > > > > > > ```python
> > > > > > > > > > > > > pred = model(Yhats_train).flatten()
> > > > > > > > > > > > > loss = MSE(pred, objectives_train)
> > > > > > > > > > > > > ```

---

### Official Review · Reviewer_NGrH · 2022-07-10

**Rating:** 5
**Confidence:** 3
**Soundness:** 3 good
**Presentation:** 4 excellent
**Contribution:** 3 good

**Summary:**

This paper proposes a novel locally optimized decision loss (LODL) for decision focused learning (DFL). The LODL loss is a parameterized function trained with the true decision loss as the target. The LODL loss is a relatively faithful measure of the decision quality, and can provide informative gradient for the DFL training.  The authors run the experiments on various optimization tasks including Linear Model, Web Advertising, and Portfolio Optimization, verifying the effectiveness of LODL. Also, the authors product some ablation studies and shows how well the LODL represents the decision quality.

**Questions:**

1. The design of LODL is interesting, but I have a concern about the complexity of fitting the local loss function.  The design fits one loss function for each sample by MSE. When the prediction space is large and needs a large number of samples, the training complexity would be very high.  The authors may discuss more about the training complexity.

2. The authors show the fitting quality of LODL by measuring the mean absolute error (MAE). From the results, we also know that the LODL may not easily fit some kinds of task loss for some inputs. It would be interesting to have some study about the relationship between the fitting quality, the gradient, and the task loss or different inputs. This may give more observations supporting the claims.

**Limitations:**

The limitations of the design is clearly listed.

**Strengths And Weaknesses:**

### Originality
The idea of training a parameterized function to approximate the decision loss is interesting. Comparing with the MSE loss in 2-stage method, the LODL is more faithful to the decision quality. Comparing with the surrogate loss in DFL, the LODL can be easier to design.

### Significance
The idea is novel for DFL.

### Quality
The proposed method is compared with multiple baselines including  2-stage, DFL, and the LODL by NN on various resource allocation problems, which verifies the advantages of LODL for some problems.

### Clarity
The paper is well-written and easy to follow.

---

> ### Author Response · Authors · 2022-08-01
> **Response to Questions**
>
> Thank you for your review! To respond to your questions:
>
> 1. **Complexity:** We broadly address this concern in the common response. However, your comment about the increasing difficulty of fitting LODLs with an increase in the dimensionality of the predicted parameters is spot on (in our experiments $\dim({y}) = 50$). We try to mitigate this issue using the localness assumptions to reduce dimensionality and having simple LODLs that are easier to fit. However, our approach is likely to be most effective for problems in which the number of parameters is low, but the cost of calling the optimization oracle is high.
> 2. **Relationship between LODL Quality and Task Loss:** We attempt to answer a version of this question in Section 5.3. There, we show that there isn't a correlation between the quality of learned LODL (according to MAE) and the Task Loss (or Decision Quality) for the set of sampled points used to train the LODL (the Gaussian Neighborhood). However, we also show that there *is* a correlation between the quality in the `Empirical Neighbourhood' (the MAE on the actual predictions that LODL encounters during training) and the Task Loss. This suggests that better LODLs could lead to improved performance. However, the bottleneck is our sampling strategy. Doing better on the sampled points does not necessarily improve performance. Harnessing improved LODL quality would require sampling more "realistic" predicted parameters somehow.
>
>     We do not compare gradients because the gradients for the Task Loss/Decision Loss can be zero almost everywhere (see the example in our response to reviewer dkZJ for more details), so matching them is generally not a good idea.

---

### Official Review · Reviewer_4d1n · 2022-07-10

**Rating:** 6
**Confidence:** 4
**Soundness:** 3 good
**Presentation:** 3 good
**Contribution:** 2 fair

**Summary:**

This paper proposes methods to approximate the decision-focused loss function that quantifies the quality of a prediction function by the quality of its induced decision. The proposed method considers several classes of locally parameterized loss functions at each label value in the sample. These loss functions are convex functions of the prediction. The parameters in the loss function for each label value in the sample are estimated by minimizing the loss approximation error at randomly sampled prediction values around the corresponding label truth.

**Questions:**

1. The proposed method learns the loss function parameters using samples generated by perturbing the label values (e.g., by adding gaussian noises). The magnitude of perturbation is obviously a very important tuning parameter. But how should this tuning parameter be selected? The selection of this tuning parameter is not discussed at all. Even the experiment section does not mention this.

2. In the WeightedMSE and DirectedWeightedMSE losses, are the weights required to be nonnegative? Line 170 to 171 suggest that the weights are free parameters without any constraints. But negative weights along some dimensions mean that large prediction errors on those dimensions can actually decrease the loss, which doesn't seem desirable. I think negative weights may also encourage predictions that are far away from the labels in the training sets, where the loss function is least accurately estimated. This may also cause problems for the idea of using locally perturbed samples.

3. The proposed approach needs to learn parameters for each label value separately. So if there are N training data points and we sample K local data points for each label value, then computing the DL loss function would require solving KN optimization problems. Isn't this very challenging for even moderately large-scale problems? Unfortunately, the computation limitation is not explicitly discussed in this paper and the problem sizes in the experiment section aren't clear about this either. For example, the perturbation process, the local sample size $K$ for the results in Table 1, and the running time information are not provided.

Some other comments:
- Is the $\hat y^\top Q\hat y$ term in portfolio optimization a type?
- The uses of the letters $n, N$ are confusing. From the equation below the line (77), $n$ refers to training data indices, and $N$ refers to training data sample size. However, in Section 5, $N$ seems to refer to the dimension of the uncertain variables, e.g.,  the number of resources or the number of stocks.
- The processes of generating data based on real datasets in web advertising and portfolio optimization experiments are not clearly described.
- Figure 2 seems interesting but I do not totally get the reason for the observed patterns. Can the authors provide some explanations?
- The authors may hope to cite some additional literature on decision-focused learning, such as
  - Elmachtoub, Adam, Jason Cheuk Nam Liang, and Ryan McNellis. "Decision trees for decision-making under the predict-then-optimize framework." International Conference on Machine Learning. PMLR, 2020.
  - Kallus, Nathan, and Xiaojie Mao. "Stochastic optimization forests." Management Science (2022).
  - Hu, Yichun, Nathan Kallus, and Xiaojie Mao. "Fast rates for contextual linear optimization." Management Science (2022).
  - Grigas, Paul, and Meng Qi. "Integrated conditional estimation-optimization." arXiv preprint arXiv:2110.12351 (2021).
  - Qi, Meng, et al. "A practical end-to-end inventory management model with deep learning." Available at SSRN 3737780 (2020).

**Limitations:**

See the comments in the box above.

**Strengths And Weaknesses:**

Originality: The idea of assigning different parameters to the loss function at different label values appears novel. This is an interesting idea to improve the expressive power of the loss function class that builds on relatively simple parametric losses (to guarantee convexity).

Quality: The writing quality of this paper is overall very good and the proposed idea in this paper is well executed. But some important limitations of the proposed approach (e.g., computation and scalability) lack discussion.

Clarity: I found this paper is overall well-written and the high-level idea of this paper is easy to grasp. One exception is that the experiment section seems to lack some important details.

Significance: Although this paper proposes an interesting idea, I am somewhat skeptical of its practicability considering the challenges in parameter tuning and the computational scalability of the proposed method.

---

> ### Author Response · Authors · 2022-08-01
> **Response to Questions**
>
> Thank you for your detailed and thoughtful comments! We will add the suggested citations and make the clarity-related changes to the camera-ready. There are additional experimental details in the appendix, but to answer your specific questions:
>
> 1. **"Selecting sampling variance"**: This is a great point and one that we have *some* intuition for. The variance has to be high enough that it leads to actually changing the decision, but low enough that it's 'realistic'. Practically, this value (along with those of other hyperparameters) is chosen via grid search over ~5 log-scaled candidates. We will update our description of the experiments to include these details.
> 2. **"Negative weights for WeightedMSE"**: This is a very astute observation! We do indeed make sure that the weights are non-negative (in fact, slightly positive) by clamping the weights to some minimum value (and initializing such that it is above that minimum value). For the 'Quadratic' and 'DirectedQuadratic' variants, we do something similar by adding some minimum amount of MSE Loss to our learned LODL. This has the effect of ensuring that the minimum eigenvalue of the learned $H$ matrix is strictly positive, i.e., you have a strictly positive curvature in all directions. We will make this more clear in the descriptions of the methods.
> 3. **"Scalability"**: We broadly address this in the common response to the reviewers above. However, this gist is that while solving $KN$ problems *is* challenging, it's a problem that is also shared by DFL. For the results in Table 1 we use 5000 samples; You can see the impact of increasing the number of samples in Table 4 in the Appendix (it also has additional details about the experimental domains!). We are also working on more concrete scalability figures and will include them in the camera-ready.
> 4. **"Figure 2"**: In this domain, the points are sampled from a uniform distribution between 0 and 1. As a result, most of the points are in the range $[-0.75, 0.75]$ and are thus downward trending. Naively trying to fit these points, as in the 2-stage setting, leads to a predictive model $M\_\theta$ that is also downward sloping. Given that $M_\theta$ has a negative slope, the predictive model thinks that the leftmost points have the highest utility and picks those (even though they \textit{actually} have the lowest utility) leading to bad outcomes. In contrast, the decision-aware models (LODL and DFL) know that having a negative slope leads to choosing bad outcomes and thus choose a positive slope (even though this does a worse job at fitting the points). The difference between the two, however, is the convexity of the surrogate; for some initializations of the predictive model, DFL is unable to determine that a negative slope is better and leads to local optima which have neither negative slope nor good model fit.

---

> > ### Comment · Reviewer_4d1n · 2022-08-10
> > **Thanks for the response**
> >
> > Thank you for your response! I appreciate that you decide to include scalability results in the camera-ready and promise to clarify the issues I mentioned. I suggest the authors can also discuss more scalability in the camera-ready. Incorporating information in the common response above will be helpful.

---

### Official Review · Reviewer_TJiM · 2022-07-11

**Rating:** 3
**Confidence:** 5
**Soundness:** 2 fair
**Presentation:** 3 good
**Contribution:** 2 fair

**Summary:**

This paper proposes LODL as a surrogate to replace the original optimization loss while approximately providing gradient information of the original optimization loss. The key argument is that the gradient information may be difficult to obtain when solving complex optimization problems, such as non-convex optimization. In LODL, a surrogate loss is constructed based on variants of MSE/quadratic functions over a dataset sampled around each training sample, such that it approximates the original optimization loss and is easily differentiable.

**Questions:**

- The computational complexity of LODL is high. It's true that we reduce the dimensionality (Line 126), but the value of $N$ can be very large (which is typically the case in decision-focused learning where the number of training samples is large). Also, we have to learn a surrogate to approximate DL for each training sample. The total complexity will be way much higher than existing methods.

- LODL is still an approximation-based method to differentiate the optimizer. There's no analytical/theoretical evidence to show that LODL can approximate the gradient of the original optimizer with respect to the predictions with sufficiently high accuracy. LODL is actually approximating DL, but this doesn't mean the gradients of DL with respect to the prediction $y$ is still well approximated by the gradient of LODL. One can easily construct counter examples in which two functions have similar values but dramatically different gradients in a neighborhood.

- The way to sample $y$ to train LODL and learn $\phi_n$ around each training sample $y_n$ is questionable. The samples are randomly generated based on a prior distribution in stage 1, but in stage 2 of learning the prediction model, the output --- predictions --- can follow a different distribution than the assumed prior distribution in stage 1. A direct consequence is that $\phi_n$ may not be accurate for the new distribution of predictions in stage 2, raising further concerns with the use of pre-trained $\phi_n$. This is also against the core idea of decision-focused learning where we want to learn the predictions by considering the entire decision pipeline as a single process.

**Limitations:**

Yes.

**Strengths And Weaknesses:**

Pros:
+ Constructing surrogates to obtain the gradients of the downstream optimization with respect to the predictions is important for decision-focused learning.

Cons:
- The computational complexity of LODL is high. It's true that we reduce the dimensionality (Line 126), but the value of $N$ can be very large (which is typically the case in decision-focused learning where the number of training samples is large). Also, we have to learn a surrogate to approximate DL for each training sample. The total complexity will be way much higher than existing methods.

- LODL is still an approximation-based method to differentiate the optimizer. There's no analytical/theoretical evidence to show that LODL can approximate the gradient of the original optimizer with respect to the predictions with sufficiently high accuracy. LODL is actually approximating DL, but this doesn't mean the gradients of DL with respect to the prediction $y$ is still well approximated by the gradient of LODL. One can easily construct counter examples in which two functions have similar values but dramatically different gradients in a neighborhood.

- The way to sample $y$ to train LODL and learn $\phi_n$ around each training sample $y_n$ is questionable. The samples are randomly generated based on a prior distribution in stage 1, but in stage 2 of learning the prediction model, the output --- predictions --- can follow a different distribution than the assumed prior distribution in stage 1. A direct consequence is that $\phi_n$ may not be accurate for the new distribution of predictions in stage 2, raising further concerns with the use of pre-trained $\phi_n$. This is also against the core idea of decision-focused learning where we want to learn the predictions by considering the entire decision pipeline as a single process.

- Some analysis of LODL would be useful, e.g., sampling complexity and generalization bounds.

- The authors are suggested to highlight the targeted scenario of LODL rather than general decision-focused learning. For example, in convex problems, we can efficiently and accurately differentiate the optimizer with respect to predictions, and so LODL is not needed or advantageous.

---

> ### Author Response · Authors · 2022-08-01
> **Response to Questions and Comments**
>
> Thank you for your detailed comments and feedback! Unfortunately, it seems like we were unable to effectively communicate the importance of surrogates for DFL in our paper and, as a result, have been unable to convince you of the value of our contribution in removing the need for such surrogates. In answering your questions, we hope to address these misunderstandings:
>
> 1. **Complexity**: We evaluate the complexity of our method and compare it to DFL in the common review above. However, the gist of the argument we make is that while our method can be expensive, DFL is *also* quite expensive. In addition, our cost can be amortized over different models whereas DFL cannot. Overall, this leads to comparable or better performance depending on the comparison regime.
> 2. **An analysis of the gradients**: For linear/discrete optimization problems (one of the major focuses of this paper), the decisions (and as a result the decision loss) are piecewise constant in the parameters. As a result, the gradients are zero almost everywhere and are not useful for training predictive models (please refer to our response to Reviewer dkZJ where we show this more concretely using an example). This is the reason why we need to develop *surrogate* optimization problems to differentiate through when using DFL. In practice, both LODL and DFL surrogates attempt to "smooth out" the decision loss in such a way that it creates useful gradients; while this is typically done by hand in the DFL, in this paper we attempt to *automate this process* by reducing it to a supervised learning problem; this is our main contribution.
> 3. **Sampling strategy**: You're right that the distributions of sampled predictions that we train our LODLs on, and the ones that they actually encounter while training the predictive model $M\_\theta$ are different. In fact, we explore this further in Section 5.3. However, there are a couple of reasons why we use this strategy:
>
>     1. *It works*: Our experiments show that we can indeed learn useful LODLs using this sampling method.
>     2. *It's cheap*: Sampling based on the actual predictions of the predictive model would be much more expensive, and likely wouldn't allow the kinds of amortization that make our method attractive from a complexity point of view.
>
>     That being said, we are currently looking into how to better sample points so that they are more aligned with the actual distribution encountered while training $M\_\theta$.
> 4. **Theoretical Analysis**: While we do not cannot ensure the similarity of the *gradients* of the Decision Loss, we do ensure that the optima of the Decision Loss and that of the LODL are the same (and are equal to the "true parameters" we're trying to learn). From a theoretical standpoint, this boils down to a "Fisher Consistency" result that we will include in the camera-ready.
> 5. **LODL Advantages**: As we highlight in our common review, the main benefits of our approach are increased *usability* rather than increased performance. That being said, even if we *can* smoothly differentiate the decision with respect to the parameters (as in some convex optimization problems), we may not want to. This is because, even though the optimization problem is *convex*, the relationship between the decision loss (of the decision produced by solving the optimization problem) and the input parameters is *non-convex*, leading to possible local optima. This is why we highlight the importance of the *convexity of the surrogates/LODLs* in this paper. We also show in the experiments that these "bad local minima" associated with non-convex surrogates lead to poor performance in practice.
>
> We hope this response helps understand the nuances and challenges associated with this problem, as well as our contributions!

---

### Author Response · Authors · 2022-08-01
**To All Reviewers**

We thank the reviewers for their thoughtful feedback! We’ve noticed some common themes in the reviews and thought that we’d respond to them here.

To start off with, we'd like to reiterate the motivation behind this paper---to make DFL **more widely usable** by avoiding the need to invent surrogate optimization problems or having to differentiate through them. The reviews evaluate our approach in the context of the literature but do not acknowledge this aspect of our contribution. In a lot of real-world contexts, the question is not "Which DFL method should I use?", but rather "Should I spend the time and effort to use DFL at all?". In those cases, our approach reduces the somewhat niche problem of differentiable optimization to one of supervised learning, which many more people are familiar with. To that end, we entreat reviewers to look beyond the specific LODL implementation used in this paper and also review our approach as a framework that allows potential users to reap the benefits of DFL without having to master Differentiable Optimization.

That being said, **complexity/scalability** is an important component of usability and something that 3 out of the 4 reviewers highlight in their review. While we did not comment on this aspect of our approach in the paper, we'd like to take the opportunity to do so here. Roughly, the amount of time taken by each of the methods is:

* 2-Stage = $\Theta(T \cdot N \cdot T\_M)$, where $T\_M$ is the amount of time taken to run one forward and backwards pass through the model ${M\_\theta}$ for one optimization instance, $N$ is the number of optimization instances, and $T$ is the number of time-steps ${M\_\theta}$ is trained for.
* DFL = $\Theta(T \cdot N \cdot (T\_M + T\_O + T'\_O))$, where $T_O$ is the time taken to solve the forward pass of one optimization instance and $T'\_O$ is the time taken to compute the backward pass.
* LODL = $\Theta(K \cdot N \cdot T\_O + N \cdot (T \cdot K \cdot T\_{LODL}) + T \cdot N \cdot T\_M)$, where $K$ is the number of samples needed to train the LODL, and $T\_{LODL}$ is the amount of time taken to run one forward and backwards pass through the LODL. The three terms correspond to (i) generating samples, (ii) training $N$ LODLs, and (iii) training $M\_\theta$ using the trained LODLs.

In practice, we find that $(T'\_O > T\_O) >> (T_M > T\_{LODL})$. As a result, the difference in complexity of DFL and LODL is roughly $\Theta(T \cdot N \cdot T\_O)$ vs. $\Theta(K \cdot N \cdot T\_O + N \cdot T \cdot K \cdot T\_{LODL})$. While our approach *can* be more computationally expensive, there are a few reasons why this typically isn't the case:

* **Simplicity of learning LODLs:** In the calculation above, we assume that LODLs are trained in the same way as $M\_\theta$. However, in practice, they can often be learned much faster, sometimes even in closed form (e.g. WeightedMSE and DirectedWeightedMSE), leading to an effective runtime of $\Theta(K \cdot N \cdot T\_O + N \cdot K \cdot T\_{LODL}) \approx \Theta(K \cdot N \cdot T\_O)$. Then, the difference between DFL and LODL boils down to $T$ vs. $K$, i.e., the number of time-steps needed to train $M\_\theta$ vs. the number of samples needed to train the LODL.
* **Amortization:** This is the biggest advantage of our approach. We need only sample candidate predictions once, to then train \textit{any number} of LODLs (e.g., WeightedMSE, DirectedQuadratic) without ever having to call an optimization oracle. Similarly, once the LODLs have been learned, you can train \textit{any number} of predictive models $M\_\theta$ based on said LODLs. In contrast, DFL requires you to call the oracle \textit{every time} you want to train a predictive model $M\_\theta$. In this sense, it is fairer to compare LODL to *meta-learning* methods, and it's easy to see that, for a large number of models to train, as is common in ML (e.g. hyperparameter/architecture search, trading-off performance vs. inference time vs. interpretability, etc.), DFL is much more expensive than our approach. *In the future, we imagine that datasets could be shipped with not only features and labels, but also LODLs associated with downstream tasks!*
* **Parallelizability:** Finally, the sample generation process for LODL is completely parallelizable, resulting in an $\Omega(T\_O)$ lower-bound wall-clock complexity for our approach. In contrast, the calls to the optimization oracle in DFL are interleaved with the training of $M_\theta$ and, as a result, cannot be parallelized with respect to $T$, resulting in an $\Omega(T \cdot T\_O)$ wall-clock complexity.

There are also additional ways in which we can speed up LODL (discussed in Section 6). Taking all of these into account, we find that our method is actually competitive with, or better than DFL in terms of scalability.

---

### Meta-Review · Area_Chair_NBjV · 2022-08-30

**Recommendation:** Accept
**Confidence:** Certain

**Metareview:**

This paper considers the problem of making decision-focused learning (DFL) more usable for both researchers and practitioners. It proposes a novel approach referred to as locally-optimized decision losses (LODL) which learns the parameters of surrogate intermediate losses to match the decision loss. Experimental results clearly demonstrate that LODL approach is able to learn effective surrogate for the considered tasks.

All the reviewers appreciated the LODL idea, but also raised a number of concerns. There was a lot of discussion and authors' have addressed most of the concerns and also acknowledged some limitations pointed out by some reviewers'. One expert reviewer who deeply engaged with the authors to both clarify and improve the paper was willing to strongly champion the paper. In their words: "It's a brilliant idea that will be foundational in the space and will be engaging and thought-provoking at the conference." Couple of reviewers' raised few points beyond the author-reviewer discussion which authors' could not see/respond to. However, I think the overall strengths of the paper outweigh these concerns.

Therefore, I recommend accepting the paper. I strongly encourage the authors' to improve the paper in terms of clarity, exposition, and additional experimental results to reflect the discussion with reviewers.

**Award:**

No

---

### Decision · Program_Chairs · 2022-09-14

Accept